

# Tall *Pinus luzmariae* trees with genes from *P. herrerae*

Christian Wehenkel[1], Samantha del Rocío Mariscal-Lucero[1,2],
M. Socorro González-Elizondo[3], Víctor A. Aguirre-Galindo[1], Matthias Fladung[4]
and Carlos A. López-Sánchez[5]

[1] Instituto de Silvicultura e Industria de la Madera, Universidad Juárez del Estado de Durango, Durango, Mexico
[2] Instituto Tecnológico del Valle del Guadiana, Tecnológico Nacional de México, Durango, Mexico
[3] CIIDIR Durango, Instituto Politécnico Nacional, Durango, Mexico
[4] Thünen-Institute of Forest Genetics, Grosshansdorf, Germany
[5] Department of Organisms and Systems Biology, University of Oviedo, Polytechnic School of Mieres, Asturias, Spain

Corresponding author
Christian Wehenkel,
wehenkel@ujed.mx

## ABSTRACT

**Context**. *Pinus herrerae* and *P. luzmariae* are endemic to western Mexico, where they cover an area of more than 1 million hectares. *Pinus herrerae* is also cultivated in field trials in South Africa and South America, because of its considerable economic importance as a source of timber and resin. Seed quality, afforestation success and desirable traits may all be influenced by the presence of hybrid trees in seed stands.

**Aims**. We aimed to determine the degree of hybridization between *P. herrerae* and *P. luzmariae* in seed stands of each species located in the Sierra Madre Occidental, Durango, Mexico.

**Methods**. AFLP molecular markers from samples of 171 trees across five populations were analyzed with STRUCTURE and NewHybrids software to determine the degree of introgressive hybridization. The accuracy of STRUCTURE and NewHybrids in detecting hybrids was quantified using the software Hybridlab 1.0. Morphological analysis of 131 samples from two populations of *P. herrerae* and two populations of *P. luzmariae* was also conducted by Random Forest classification. The data were compared by Principal Coordinate Analysis (PCoA) in GenAlex 6.501.

**Results**. Hybridization between *Pinus herrerae* and *P. luzmariae* was observed in all seed stands under study and resulted in enhancement of desirable silvicultural traits in the latter species. In *P. luzmariae*, only about 16% molecularly detected hybrids correspond to those identified on a morphological basis. However, the morphology of *P. herrerae* is not consistent with the molecularly identified hybrids from one population and is only consistent with 3.3 of those from the other population.

**Conclusions**. This is the first report of hybrid vigour (heterosis) in Mexican pines. Information about hybridization and introgression is essential for developing effective future breeding programs, successful establishment of plantations and management of natural forest stands. Understanding how natural hybridization may influence the evolution and adaptation of pines to climate change is a cornerstone to sustainable forest management including adaptive silviculture.

## INTRODUCTION

Hybridization represents an important evolutionary force that can introduce much more new genetic material than is created by mutation events (*Anderson, 1949*; *Wright, 1964*). It can also act as an additional, perhaps more abundant, source of adaptive genetic variation than mutation (*Grant & Grant, 1994*), by allowing gene flow and recombination (*Abbott et al., 2013*; *Hipp, 2018*). Furthermore, hybridization is one of the key sources of species formation and diversity, and many species may have originated by this route (*Linder & Risenberg, 2004*; *Blanckaert & Bank, 2018*), perhaps even as much as between 30% and 80% of all species (*Wendel, McD & Rettig, 1991*). On the other hand, increasing rates of hybridization may also lead to the extinction of unique populations or species because of unsuccessful reproductive efforts or introgression with a more common species (*Rhymer & Simberloff, 1996*; *Blanckaert & Bank, 2018*). In times of rapid ongoing climate change, hybridization may thus contribute to further extinctions, sometimes weakening reproductive isolation among species (*Owens & Samuk, 2019*) as well as supporting the development of novel segregating genotypes that will speed up adaptation to changes in climate (*Chunco, 2014*; *Menon et al., 2019*). Knowledge of hybridization has therefore deep practical reasons. Besides the adaptation issues, the presence of hybrid trees in seed stands "contaminates" the species' gene pool and thus may influence seed quality and afforestation success (*Arnold & Hodges, 1995*; *Rieseberg & Carney, 1998*).

The process of hybridization incorporates alleles from one species into the gene pool of another (*Harrison, 1993*). Interactions between the environment and genetic structure can thus lead to segregation of a novel taxon from parental types. Depending on the degree of differentiation, hybrid offspring of two or more plants of different taxa are sometimes identified as species, subspecies or variants (*Futuyma, 1998*; *Tovar-Sánchez & Oyama, 2004*).

Hybrids often display post-mating reproductive difficulties relative to their ancestors. These difficulties include hybrid weakness, sterility and fitness breakdown (*Rieseberg & Carney, 1998*). However, hybrids are not necessarily uniformly unfit. On the contrary, some genotypic classes may be equally fit or even fitter than the parental taxa (*Arnold & Hodges, 1995*; *Mabaso, Ham & Nel, 2019*). The first hybrid generation ($F_1$) tends to exceed the parental generation in vegetative vigour or robustness, in a condition also known as heterosis. However, early hybrid generations such as $F_2$ and $F_3$ are often less vigorous and fertile than their parents due to the break-up of adaptive gene arrangements (*Rieseberg & Carney, 1998*).

Studies involving hybridization are often based on morphological traits. However, the phenotypic expression of characters of one taxon in another does not necessarily indicate hybridization. Similar characters may occur in species because of phenotypic plasticity, convergent evolution or simply because of a common ancestry, as *Linder et al. (1998)* observed in wild sunflower. Furthermore, morphological characters yield limited information when the parents and their hybrids are affected by environmental factors such as disease or drought stress, generating a wide range of phenotypic variability. This problem is increased by subsequent backcrossing of the hybrids to either parent species, resulting

in morphological characters that become more similar to those of the backcrossed parent species (*Chen et al., 2004*).

Use of molecular markers to detect interspecific hybridization is more effective than verification by morphological, chemical or cytogenetic analysis, especially as access is available to an almost unlimited number of molecular markers (*Rieseberg, Ellstrand & Arnold, 1993*; *Alexandrov & Karlov, 2018*; *Jasso-Martínez et al., 2018*). Introgressive hybridization in many plant species has been identified by molecular data (*Rieseberg, Ellstrand & Arnold, 1993*; *Arnold, 1997*; *Delgado et al., 2007*; *Kaplan & Fehrer, 2007*; *McVay, Hipp & Manos, 2017*). These markers have been useful for diagnosing $F_1$ and derived hybrid generations, evaluating levels of gene flow among species and reconstructing phylogenetic relationships between hybridizing taxa and their close relatives (*Rieseberg, Ellstrand & Arnold, 1993*).

Amplified Fragment Length Polymorphism (AFLP) markers have been successfully used to detect introgressive hybridization in plants (*Guo et al., 2005*; *Shasany et al., 2005*; *Koerber, Anderson & Seekamp, 2013*), specifically in pines (*Xu, Tauer & Nelson, 2008*; *Stewart et al., 2010*; *Vasilyeva & Semerikov, 2014*; *Ávila Flores et al., 2016*). AFLP markers include a more or less large number of polymorphic, di-allelic loci and can be developed relatively easily and at a relatively low cost, even for species about which no prior genetic information is available (*Mueller & Wolfenbarger, 1999*; *Hardy et al., 2003*; *Paun & Schönswetter, 2012*). Possible disadvantages of the AFLP technique such as compiling standardized patterns in a database for interlaboratory use and future reference can be avoided by using specific procedures as recommended by *Savelkoul et al. (1999)*. However, AFLP as dominant marker does not allow identification of homologous alleles and thus scoring of homozygote and heterozygote states (*Mueller & Wolfenbarger, 1999*).

Interspecific hybridization is very common in natural stands of the genus *Pinus* (*Critchfield, 1967*; *Quijada et al., 1997*; *Conkle & Critchfield, 1988*; *López-Upton et al., 2001*; *Delgado et al., 2007*; *Ávila Flores et al., 2016*; *Stacy et al., 2017*; *Vasilyeva & Goroshkevich, 2018*; *Mo et al., 2019*), because of very weak reproductive barriers between pine species (*Little & Righter, 1965*; *Garrett, 1979*; *Dungey, 2001*); this could be generalized across conifers with similar divergence history (*Menon et al., 2018*). Interspecific $F_1$ hybrids in this genus are highly viable and fertile (*Critchfield, 1975*), which complicates taxonomic classification (*Martínez, 1948*; *Lanner, 1974*). Genetic diversity is often high in *Pinus* because of the usually large populations, cross-fertilization, high mutation rates and long-distance dispersion of pollen and sometimes seeds (*Gernandt et al., 2011*), as well as interspecific hybridization and introgression (*Critchfield, 1967*; *Critchfield, 1975*; *Quijada et al., 1997*; *Conkle & Critchfield, 1988*; *Ledig, 1998*; *López-Upton et al., 2001*; *Ávila Flores et al., 2016*). In addition, understanding the phylogenetic relationships between closely related species of pines is also difficult due to retention of ancestral alleles (*Delgado et al., 2007*; *Hernández-León et al., 2013*; *Ortiz-Martínez & Gernandt, 2016*). Moreover, North American hard pines in the subsection *Australes* share plastid DNA lineages due to introgressive hybridization or incomplete lineage sorting (*Ortiz-Martínez & Gernandt, 2016*).

*Pinus herrerae* Martínez and *Pinus luzmariae* Pérez de la Rosa belong to the subsection *Australes*, a monophyletic group including 29 pine species (*Gernandt et al., 2005*;

*Hernández-León et al., 2013*). Herrera's pine (*P. herrerae*), previously known as *Pinus teocote* var. *herrerae* (Martínez) Silba, is endemic to western Mexico, where it covers an area of about 1 million hectares (1 M ha) (*Comisión Nacional Forestal, 2009*) in mountain ranges between 16° and 28°N, at elevations ranging from 1,100 m to 2,800 m (*Dvorak et al., 2007*; *Wehenkel et al., 2015*). The species is used to produce construction timber and resin (*Martínez, 1948*). It is also cultivated as an exotic in field trials in South Africa and South America because of its typically very tall, straight trunk (*Dvorak et al., 2007*). *Pinus luzmariae* (three-needled egg-cone pine), previously known as *Pinus oocarpa* var. *trifoliata* Martínez, was first recognized as a separate species by *Pérez de la Rosa (1998)*. This small to medium-sized tree species is endemic to Mexico and it has been reported as covering an area of about 200,000 ha (*Comisión Nacional Forestal, 2009*). However, its distribution is not clear because it has been included in the very wide range of *P. oocarpa* Schiede ex Schltdl. The two largest populations are documented in the southern Sierra Madre Occidental covering about 1,000 hectares in south Durango and about 600 ha in northern Jalisco, respectively. Although no uses have been documented for *Pinus luzmariae*, it may be used as a source of timber, in the same way as *P. oocarpa*. The number of mature individuals of this species in its natural habitat is decreasing (*Pérez de la Rosa & Farjon, 2013*).

Both pine species grow in the Madrean-tropical subregion of the Sierra Madre Occidental, at lower elevations (< 2,400 m). *Pinus herrerae* often is dominating in subhumid areas whereas *P. luzmariae* occupies sites with poor soils, although sometimes they grow together (*González-Elizondo et al., 2012*; *González-Elizondo et al., 2013*). The ecological niches of these two species are clearly defined by soil pH and climate in the State of Durango (Mexico) (*Wehenkel et al., 2015*).

Population genetics studies of *P. herrerae* are scarce (see *Wehenkel et al., 2015*) and of *P. luzmariae* non-existent. Hybrids between these two species have not yet been reported so far. The aim of the present study was therefore to use AFLP molecular markers and morphological traits to determine the degree of hybridization between *P. herrerae* and *P. luzmariae* in seed stands of each species located in the Sierra Madre Occidental mountain system, Durango, Mexico. Although *P. herrerae* and *P. luzmariae* are morphologically very different (*Perry, 1991*; *Farjon & Styles, 1997*; *García-Arévalo & González-Elizondo, 2003*; *Pérez de la Rosa & Vargas Amado, 2009*), they are genetically closely related and can thus, theoretically, easily hybridize with each other (*Dvorak et al., 2000*; *Ortiz-Martínez & Gernandt, 2016*; *Gernandt et al., 2018*). In addition, we tested the possible *P. luzmariae* hybrid individuals for clues of possible hybrid vigour (heterosis). We aimed to unravel introgressive hybridization between *P. herrerae* and *P. luzmariae*, under the assumption that effective pollen flow has occurred between the two species.

## MATERIAL AND METHODS

### Study sites

Samples were obtained from trees grown in three *Pinus herrerae* (PH) and two *P. luzmariae* (PL) seed stands located in the Sierra Madre Occidental, state of Durango (NW Mexico) (collection permit SEMARNAT SGPA/DGVS/003644/18). The three *P. herrerae* seed stands

**Table 1  Locations of the the stands of *Pinus herrerae* (PH) and *Pinus luzmariae* (PL) under study.**

| Abbreviated stand name | Property | Seed stand | Municipality | Latitude (N) | Longitude (W) | Elevation (m) |
|---|---|---|---|---|---|---|
| PH-R | Comunidad Milpillas | Ranchito | Pueblo Nuevo | 23° 31′46.8″ | 105° 05′11.3′ | 2,511 |
| PH-A | Comunidad Lajas | Manchon del Abies | Pueblo Nuevo | 23° 11′15.1″ | 105° 02′45.5′ | 2,318 |
| PH-V | Comunidad Lajas | Ventana | Pueblo Nuevo | 23° 12′08.3″ | 105° 01′13.7″ | 2,396 |
| PL-L | Comunidad Lajas | Laguna | Pueblo Nuevo | 23° 10′18.4″ | 105° 07′25.4′ | 1,960 |
| PL-T | Comunidad Lajas | Tacuache | Pueblo Nuevo | 23° 10′47.8″ | 105° 08′46.0″ | 2,140 |

were (1) Ranchito (PH-R), (2) Manchón del Abies (PH-A) and (3) Ventana (PH-V). The *P. luzmariae* stands were (4) Laguna (PL-L) and (5) Tacuache (PL-T). All seed stands are uneven-aged and located in natural populations (Table 1).

The three PH seed stands grow on slightly acidic soil (pH 5.2 ± 0.4 (standard deviation [SD]), with $H^+$ representing 27.4 ± 6.2 SD of total exchangeable cations) (*Wehenkel et al., 2015*). The Julian date of the last frost date in spring ($S_{day}$) was 118 (equivalent to April 28) ±13 days SD. The elevation ranges between 2,318 and 2,511 m above sea level in the study area, with annual rainfall between 1,046 and 1,116 mm. The mean temperature varies from about 11 to 13 °C. The PL stands are also situated on slightly acidic soils with pH 5.0 ± 0.4 SD and $H^+$ representing 29.3 ± 5.6% SD of total exchangeable cations, although at lower elevations with an earlier $S_{day}$ and higher temperatures. Their elevation varies from 1,960 to 2,140 m above sea level, $S_{day}$ is 77 (equivalent to March 18 ± 13 days SD, with annual rainfall of between 1,107 and 1,139 mm. The mean temperature ranges between 14 and 16 °C.

The PH-R and PH-A stands are separated from PL-L and PL-T by a deep (1,400 m) canyon and by a distance of 8.1–11 km (Fig. 1). The three PH stands include typical specimens of *P. herrerae*, i.e., tall trees, of height up to 40 m. However, both populations of *P. luzmariae* under study showed uncommon increased fitness relative to other populations of the same species (e.g., *Pérez de la Rosa, 1998*; *García-Arévalo & González-Elizondo, 2003*), as they are taller (see more in 'Discussion').

## Fluorescence-based semi-automated AFLP analysis

Needles were collected from a total of 171 adult, dominant and superior putative phenotypes, i.e., plus trees, according to previously described selection criteria (*Wehenkel et al., 2015*), of both *P. herrerae* and *P. luzmariae* (33–35 per stand). Dendrometric variables were also recorded in all seed stands, including coordinates, height ($H$) and diameter at breast height ($DBH$) of each sampled tree. The samples were placed in individual tubes with a few drops of ethanol and stored at −10 °C until DNA extraction.

DNA was extracted using the QIAGEN DNeasy96 plant kit, according to the steps described in the product manual. DNA fingerprints were obtained by amplified fragment length polymorphism (AFLP), according to a modification of the protocol of *Vos et al. (1995)*, outlined by *Ávila Flores et al. (2016)*. The restriction enzymes used were *Eco*RI

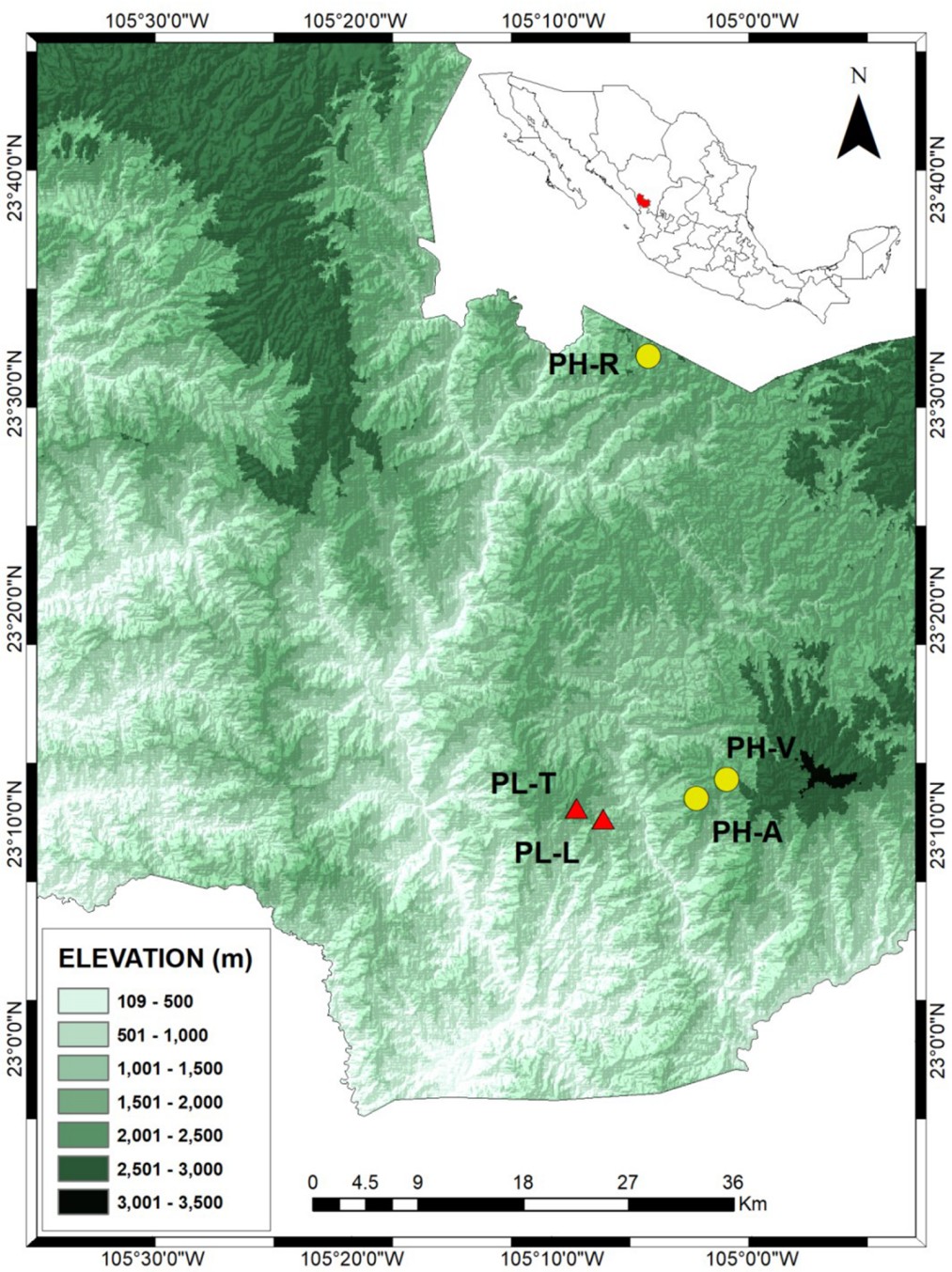

**Figure 1** **Locations of the *Pinus herrerae* (yellow circles) and *Pinus luzmariae* stands (red triangles) in the State of Durango, Northwest Mexico.** Locations of the *Pinus herrerae* (*Pinus teocote* var. *herrerae*) (yellow circles) and *Pinus luzmariae* stands (red triangles) in the State of Durango, Northwest Mexico. The *P. herrerae* seed stands were 1) Ranchito (PH-R), Manchon del Abies (PH-A) and Ventana (PH-V). The *P. luzmariae* stands were Laguna (PL-L) and Tacuache (PL-T).

(selective primer: 5′-GACTGCGTACCAATTCNNN-3′) and *Mse*I (selective primer: 5′-GATGAGTCCTGAGTAANNN-3′). The primer combination E01/M03 (*Eco*RI-A/*Mse*I-G) was used in the pre-AFLP amplification.

Selective amplification was carried out with the fluorescent-labelled (FAM) primer pair E35 (*Eco*RI-ACA-3) and M63+C (*Mse*I-GAAC). All PCR reactions were carried out in a Peltier Thermal Cycler (MJ Research, Waltham, Massachusetts, USA). The amplified restriction products were electrophoretically separated in a Genetic Analyzer (ABI 3100 16 capillaries), with a GeneScan 500 ROX internal size standard (Applied Biosystems, Foster City, California, USA). The size of the AFLP fragments was resolved with the GeneScan® 3.7 and Genotyper® 3.7 software packages (Applied Biosystems, Foster City, California, USA).

The amplified restriction products were scored automatically. Only high quality fragments above the signal threshold of 50 (minimum peak height) (according to ABI manual) and with a maximum peak width of 1.0, minimum fragment size of 75 base pairs (bp), maximum fragment size of 450 bp and tolerance +/- bp of 0.4 were considered. Two fragments were only considered when the peak-peak distance between the two signals was at least 0.5 bp. The quality and reproducibility of the analysis were verified by inclusion of reference samples in each plate and independent repetition (replicate PCRs) of analysis at least 16 samples (i.e., a minimum of 16 randomly chosen individuals from each plate). In all replicates, the AFLP pattern was the same as in the first analysis (*Simental-Rodríguez et al., 2014*; *Ávila Flores et al., 2016*).

Two binary AFLP matrices were generated from the presence (code 1) or absence (code 0) at probable band positions (Table S1). The bands detected represented the presence of a dominant genetic variant (plus phenotype) with unknown mode of inheritance of this band position (detected fragment length) (*Vuylsteke et al., 1999*; *Krauss, 2000*). The absence of a band indicated the presence of only recessive genetic (allelic) variants at the given position (locus). To minimize the rate of size homoplasy (*Vekemans & Hardy, 2004*; *Caballero, Quesada & Rolán-Alvarez, 2008*) and technical artefacts (*Krauss, 2000*), only the polymorphic loci (fragment lengths) with frequencies of occurrence of between 5 and 95% were selected for study (*SanCristobal et al., 2006*).

## Defining pure individuals and molecular identification of hybrids

The trees PH-V4, PH-V49, PH-V52, PH-V64 and PH-V127, and PL-T28, PL-T31, PL-T37, PL-T103, PL-T130 were defined as individuals of "pure" *Pinus herrerae* (PH) and *P. luzmariae* (PL), respectively, (hereinafter called pure individuals or pure trees) identified by their genetic affiliation probability and by their morphological traits (see details below).

When PH or PL stands include common hybrid trees, they should possess a genome that is a combination of alleles derived from trees belonging to both species. These hybrids can be detected by genetic data obtained from molecular marker analysis (*Xu, Tauer & Nelson, 2008*; *Ávila Flores et al., 2016*).

The resulting AFLP loci from the 171 tree samples were used to determine the degree of introgressive hybridization between PH and PL in the analysis, conducted with STRUCTURE version 2.1 (*Pritchard, Stephens & Donnelly, 2000*; *Falush, Stephens &*

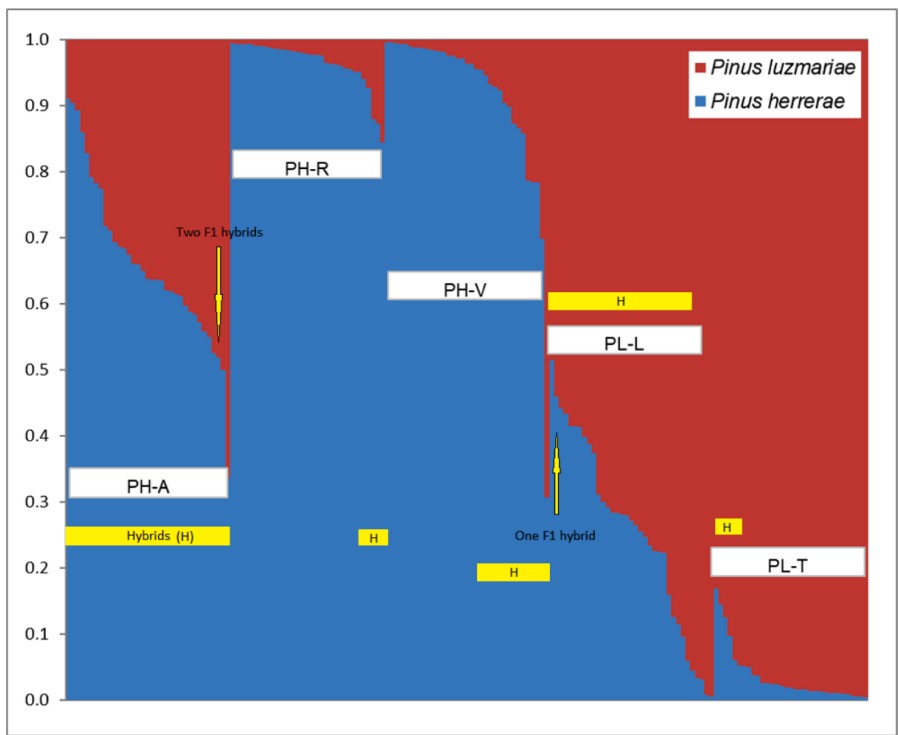

**Figure 2** Identification of two populations ($K = 2$) based on 348 AFLP from three *Pinus herrerae* seed stands (PH) (Pop 1 = blue) and two *Pinus luzmariae* seed stands (PL) (Pop 2 = orange) using STRUC-TURE. Identification of two populations ($K = 2$) based on AFLP data from three *Pinus herrerae* seed stands (PH) (Pop 1 = blue) and two *Pinus luzmariae* seed stands (PL) (Pop 2 = red) (171 individuals in total), with Structure, version 2.1 software. PH-A with 35 hybrids, PH-R with six hybrids, PH-V with 14 hybrids, PL-L with 30 hybrids and PL-T with seven hybrids; PH-A, Manchon del Abies; PH-R, Ranchito; PH-V, Ventana; PL-L, Laguna and PL-T, Tacuache.

*Pritchard, 2007*) and NewHybrids version 1.1 Beta 3 software (*Anderson & Thompson, 2002*). Both software programs have been used to identify putative hybrids in *Pinus* with dominant markers such as AFLP (*Xu, Tauer & Nelson, 2008*; *Ávila Flores et al., 2016*). The systematic Bayesian clustering approach applying Markov Chain Monte Carlo (MCMC) estimation, as implemented in STRUCTURE was used to test the affiliation of individuals to each species. The MCMC process started by randomly assigning individuals to a pre-determined population (group or species) number ($K$) (here $K = 2$, Fig. 2). Repeated many times in the burn-in process (burn-in period of 10,000 cycles), comprising 100,000 iterations, variant (allele) frequencies were estimated in each population and individuals re-assigned using those frequency estimates. In the course of the process, the convergence progressed toward reliable membership probabilities of individuals to a population (or species) (*Porras-Hurtado et al., 2013*).

If the probability of PH (or PL) affiliation of a putative PH (or PL) tree was less than 95% according to STRUCTURE, then this individual was recorded as a candidate hybrid. The affiliation probability was measured by the proportion of the dominant STRUCTURE populations in the studied stands (Table S2). Individuals were identified as first-generation

($F_1$) hybrids when the probability of PH affiliation with a PL tree was in the range 48–52% (*Xu, Tauer & Nelson, 2008*; *Ávila Flores et al., 2016*).

Use of the Markov chain Monte Carlo (MCMC) methodology and 100,000 sweeps after BurnIn (10,000 cycles), the NewHybrids 1.1 software (*Anderson & Thompson, 2002*) is suitable for the situation studied here, where only two diploid species appear to be hybridizing. By applying this software, *Anderson (2008)* showed that just ten AFLP were adequate to accurately separate parental and $F_1$ genotypes from later generation hybrid classes. A sample of $M$ individuals, putative pure individuals as well as hybrid individuals, is obtained and genotyped at the $L$ loci of codominant and dominant genetic markers. This software contemplates six genotype classifications (pure species 1, pure species 2, $F_1$ hybrids, $F_2$ hybrids, and the first backcross generation to pure species 1 or pure species 2) and estimates the probability that each individual belongs to the different classes (*Anderson & Thompson, 2002*; *Anderson, 2008*; *Xu, Tauer & Nelson, 2008*) (Table S3). A tree was assigned to one of the hybrid classes with a posterior probability of at least 95%.

To visualize individual and species differences, Principal Coordinate Analysis (PCoA) was also performed using the binary AFLP data matrix produced, Nei's Genetic Distance (*Nei, 1972*; *Nei, 1978*) and GenAlex 6.501 software (*Peakall & Smouse, 2012*). The PCoA diagrams were elaborated with the first, second and third coordinate.

The accuracy of the software STRUCTURE and NewHybrids (burn-in period of 10,000 cycles, 100,000 iterations) in detecting hybrids was quantified using the computer program Hybridlab 1.0 (*Nielsen, Bach & Kotlicki, 2006*). However, this software simulates intraspecific hybrids from population samples of co-dominant nuclear genetic markers, whereas the AFLP-technique can detect only dominant genetic markers. Here, the accuracy corresponds with the number of correctly identified individuals for a hybrid generation over the actual number of individuals assigned to that generation (*Marie, Bernatchez & Garant, 2011*).

Assuming that the fixed band differences between PH and PL were homozygous (expected for fixed polymorphisms), a subset of 11 diagnostic AFLP loci of the five pure PH and PL trees distinguishing the two parental species (100% of one reference parental species had the band whereas 0% of the other parental species did not) were used to simulate three intraspecific hybrid generations ($F_1$ ($N = 50$) and $F_2$, $F_1$PL, $F_1$PH, $F_2$PL, $F_2$PH, $F_1$PL-PL and $F_1$PH-PH backcrosses; $N = 125$ for each). The majority of the used AFLP loci did not show fixed band differences between PH and PL. Consequently, it was not possible to reliably identify the heterozygote or homozygote state by means of the AFLP bands, as found also by *LaRue, Grimm & Thum (2013)*. Nevertheless, these three simulated intraspecific hybrid generations were also created when using all polymorphic AFLP loci assuming that the AFLP band was always the dominant homozygote and the recessive variant the recessive homozygote. Finally, we performed STRUCTURE and NewHybrids analyses to estimate their accuracy using the two simulation datasets.

**Morphological detection of hybrids**
To test the results of the AFLP analysis, morphological analysis was conducted on samples from the same trees in populations PH-A, PH-V, PL-L and PL-T sampled for the AFLP

**Table 2   Cone and needle traits of pure *Pinus herrerae* and *Pinus luzmariae* used in the study.**

| Morphological traits | *Pinus herrerae* | | | *Pinus luzmariae* | | |
|---|---|---|---|---|---|---|
| | max | mean | min | max | mean | min |
| **Cone shape** (ovoid (1) vs. widely ovoid (2)) | 1 | 1 | 1 | 2 | 2 | 2 |
| **Cone width (cm)** | 3.5 | 3.1 | 2.8 | 5.2 | 4.8 | 4.1 |
| **Cone scale position** (ascendant (1) vs. divergent (3)) | 1 | 1 | 1 | 2 | 2 | 2 |
| **Cone scale length (cm)** | 1.5 | 1.3 | 1.1 | 1.9 | 1.7 | 1.7 |
| **Cone scale width (cm)** | 0.8 | 0.7 | 0.6 | 1.1 | 0.9 | 0.8 |
| **Apophysis width (mm)** | 5 | 4.6 | 4 | 8 | 6.8 | 6 |
| **Keel** (inconspicuos (0) vs. prominent (1)) | 1 | 0.4 | 0 | 0 | 0 | 0 |
| **Leaf sheath length (cm)** | 1.3 | 1.1 | 0.9 | 2.0 | 1.8 | 1.5 |
| **Leaf sheath diameter (mm)** | 1.4 | 1.2 | 1.0 | 1.9 | 1.8 | 1.7 |
| **Needle number** | 3.0 | 3.0 | 3.0 | 3.5 | 3.2 | 3.0 |
| **Needle length (cm)** | 15.9 | 13.4 | 11.3 | 28.8 | 24.4 | 21.5 |
| **Needle width (mm)** | 1.0 | 0.8 | 0.7 | 1.3 | 1.2 | 1.1 |
| **Needle thickness (mm)** | 0.3 | 0.2 | 0.1 | 0.6 | 0.5 | 0.4 |
| **Stomata rows (dorsal face)** | 9.0 | 7.1 | 5.3 | 10.0 | 9.7 | 9.0 |
| **Stomata rows (ventral faces)** | 3.3 | 3.1 | 3.0 | 6.0 | 5.0 | 4.0 |

analysis. At least 31 individual trees were analysed for cone traits and 11 for needle traits per species. Samples of branchlets, needles and cones were collected for taxonomic determination and morphometric examination, and voucher specimens were deposited in the CIIDIR herbarium (acronym according to *Thiers, 2019*), the collection of the Centro Interdisciplinario de Investigación para el Desarrollo Integral Regional of the Instituto Politécnico Nacional (Tables S4 and S5). Morphological characters were selected from those used by *Martínez (1948)* and *Pérez de la Rosa (1998)* in their descriptions of *P. herrerae* and *P. luzmariae*, respectively, as well as those used by *García-Arévalo & González-Elizondo (2003)* to distinguish these two species in the study area, considering sheath, needle and cone characters (Table 2, Tables S6 and S7, Fig. S1). Characters that do not possess discrete or different states between *P. herrerae* and *P. luzmariae* according to these authors were excluded from the analysis as they have no informative value for this study, e.g., persistence of fascicle sheaths (persistent in both), and cone peduncle (peduncle present, oblique and about the same diameter in both species). Some of the characters were measured at ×40 with the aid of a Carl Zeiss Dicovery.V8 stereo microscope. Individual and species differences were pictured by PCoA using the distance of *Huff, Peakall & Smouse (1993)* and GenAlex 6.501.

In order to detect PH and PL hybrids identified by morphological traits, we first identified five pure PH (PH-V 4, 49, 52, 64, 127) and five pure PL (PL-T 28, 31, 37, 103, 130) trees, applying a genetic affiliation probability larger than 0.99 according to STRUCTURE and NewHybids and clearly assignable by morphological traits. Since not every independent morphological trait was normally distributed and continuous, the species assignment

of each tree and "morphological" hybrids was established by Random Forest (*Breiman, 2001*) using the caret package and function "train" (*Kuhn, 2008*; *Williams et al., 2018*) available in the free statistical application R 3.5.2 (*R Development Core Team, 2018*). For this purpose, the PH trees were labeled with the value "1" (corresponding to the presence of PH) and the PL trees were labelled with "0" (corresponding to the absence of PH) in this presence–absence classification model. The model for both the cone traits and needle traits, respectively, was fit using a 5-fold cross-validation repeated 10 times (i.e., using 80% of the dataset as training set and the remaining 20% as testing set). Random Forest is a nonparametric tree-based classifier and hence does not require variable scaling and can successfully handle non-normality (*Strobl, Malley & Tutz, 2009*) as well as categorical and confounding variables (*Dormann et al., 2013*). Caret package (short for Classification And REgression Training) is a complete framework for building machine learning models (http://caret.r-forge.r-project.org).

Using all morphological traits listed in Table 2, a tree was classified as a possible PH (or PL) tree if it was assigned to each of those species with the highest assignment probability (i.e., > 50%). If the posterior probability of PH (or PL) affiliation of a possible PH (or PL) tree was less than 95% according to Random Forest, then, that tree was considered as a putative PH (or PL) hybrid.

The predictive ability of the Random Forest model was evaluated using the True Skill Statistic (*TSS*; *Allouche, Tsoar & Kadmon, 2006*) using the caret package in *R* (for more details see *Escobar-Flores et al., 2018*). *TSS* (also known as the Hanssen–Kuipers discriminant) is an appropriate alternative to Area Under a Receiver Operating Characteristic (*ROC*) Curve (*AUC*; *Fawcett, 2006*) in cases where model predictions are formulated as presence–absence models and an improvement to the widely used kappa. *TSS* not only accounts for both omission and commission errors, but is not affected by the sample size of each class. The *TSS* is defined as sensitivity + specificity −1, and ranges from −1 to +1, where +1 indicates a perfect classification model and values of zero or less indicate performance no better than random (*Allouche, Tsoar & Kadmon, 2006*; *Tatler, Cassey & Prowse, 2018*).

## RESULTS

### Molecular detection of hybrids

The AFLP primer combination yielded 348 polymorphic bands of 75-450 base pairs across all individual specimens of *Pinus herrerae* (PH) and *Pinus luzmariae* (PL). PH yielded 338 and PL 316 polymorphic bands. Both species shared 304 AFLP fragments (87.4% of polymorphic bands detected).

Figure 2 shows the percentage of hybridization obtained with the STRUCTURE software, for $K = 2$, the three PH seed stands have a dominant genetic variant (blue) and the two PL seed stands contain another dominant genetic variant (red). Based on a 5% probability of introgression of gene content, 92 (53.8%) putative hybrids between PH and PL were found in all the seed stands analysed. Thus, 18% of the individuals in the Ranchito PH stand (PH-R) were putative hybrids; all PH individuals in the Manchón del Abies PH

**Table 3** Detection of hybrid trees by analysis of 348 AFLP markers with the software programs STRUCTURE version 2.1 and NewHybrids version 1.1.

| | | STRUCTURE 2.1 | | NewHybrids 1.1 | | |
|---|---|---|---|---|---|---|
| Seed stand | Sample number | Hybrid number | $F_1$ hybrid number | Hybrid number | $F_1$ hybrid number | Backcrossing number |
| PH-A | 35 | 35 | 2 | 9 | 0 | 0 |
| PH-R | 33 | 6 | 0 | 0 | 0 | 0 |
| PH-V | 35 | 14 | 0 | 2 | 0 | 1 |
| PL-L | 35 | 30 | 1 | 33 | 0 | 14 |
| PL-T | 33 | 7 | 0 | 21 | 0 | 4 |
| Total | | 92 | 3 | 65 | 0 | 19 |

**Notes.**

PH, *Pinus herrerae*; PL, *Pinus luzmariae* seed stands; PH-A, Manchon del Abies; PH-R, Ranchito; PH-V, Ventana; PL-L, Laguna; PL-T, Tacuache.

stand (PH-A) displayed genetic introgression from PL, and 14 of the 35 individuals in the Ventana PH stand (PH-V) were putative hybrids (40.0%). Regarding *P. luzmariae*, 30 (85.7%) putative hybrids were detected in the Laguna stand (PL-L), whereas seven individuals (21.2%) in the Tacuache *P. luzmariae* stand (PL-T) displayed introgression with *P. herrerae*. Five trees were first-generation hybrids ($F_1$), as indicated by introgression of between 48 and 52%: three trees in the PH-A stand and another two in the PL-L stand (Fig. 2, Table 3).

NewHybrids software clearly identified 65 (38%) putative hybrids between PH and PL (Table 3). No putative hybrids were found in the Ranchito PH stand (PH-R). In total, 25.7% of the individuals in the PH-A were putative hybrids, and two of the 35 individuals in the Ventana PH stand (PH-V) were putative hybrids (5.7%). A large majority (94.2%) of the individuals in the Laguna *P. luzmariae* stand (PL-L) were identified as putative hybrids, whereas 64% of the individuals in the Tacuache *P. luzmariae* seed stand (PL-T) displayed genetic introgression with *P. herrerae*. Only one tree, located in the PL-L stand, was detected as a first-generation hybrid ($F_1$) (Table 3).

The accuracy test showed that the method NewHybrids (NH) correctly assigned at least 88% of naturally occurring "pure" PH and PL individuals using 11 diagnostic AFLP and all 348 AFLP. STRUCTURE (STR) presented much more errors, especially with the 11 diagnostic AFLP. Using the 11 diagnostic loci, for both methods detections of 1st and 2nd ($F_2$, $F_1 PL$ and $F_1 PH$ backcrosses) generation hybrids were 100% and nearly 100%, for 3rd generation hybrids ($F_2 PL$, $F_2 PH$, $F_1 PL$-PL and $F_1 PH$-PH backcrosses) this decreased further to 0.59% in STR and 0.49% in NH (posterior probability (PP) of at least 95%). Using the all 348 AFLP, simulations demonstrated lower rates of inaccurately than the test with 11 diagnostic loci. STR and NH correctly assigned 100% of simulated $F_1$ hybrids and nearly 100% of 2nd generation hybrids. Moreover, NH correctly assigned 100% of simulated 3rd generation hybrids, too. Using STR, a lower percentage of 3rd generation hybrids were correctly assigned (50%) (PP of at least 95%) (Table 4).

**Table 4** Accuracy of assignment of *Pinus herrerae* (PH), *Pinus luzmariae* (PL) and their hybrids using STRUCTURE 2.1 (STR, *Pritchard, Stephens & Donnelly, 2000*; *Falush, Stephens & Pritchard, 2007*) and NewHybrids 1.1 (NH, *Anderson & Thompson, 2002*) using a subset of 11 diagnostic AFLP loci and all 348 polymorphic AFLP loci found in the study. Hybrid classes are as follows: 1st gen— $F_1$, 2nd gen— $F_2$ and $F_1$ backcrosses, and 3rd gen— $F_2$ backcrosses and $F_1$ double backcrosses (*Cullingham et al., 2011*).

| Class | 11 AFLP loci | | 348 AFLP loci | |
|---|---|---|---|---|
| | STR | NH | STR | NH |
| 1st gen | 1 | 1 | 1 | 1 |
| 2nd gen | 1 | 0.99 | 1 | 0.99 |
| 3rd gen | 0.59 | 0.49 | 0.50 | 1.00 |
| Hybrid Avg. | **0.86** | **0.83** | **0.83** | **1.00** |
| PH | 0.13 | 0.88 | 0.80 | 0.88 |
| PL | 0.13 | 0.88 | 0.78 | 0.95 |

**Table 5** Detection of hybrid trees (*Pinus herrerae* × *Pinus luzmariae*) by analysis of seven cone and eight needle traits using Random Forest classification.

| Seed stand | Cone traits | | Needle traits | |
|---|---|---|---|---|
| | Sample number | Hybrid number | Sample number | Hybrid number |
| PH-A | 35 | 3(7) | 34 | 0(2) |
| PH-V | 34 | 0(10) | 31 | 2(3) |
| PL-L | 31 | 1(3) | 13 | 0(4) |
| PL-T | 31 | 2(9) | 11 | 2(3) |
| Total | 131 | 6(29) | 89 | 4(12) |

**Notes.**

PH, *Pinus herrerae*; PL, *Pinus luzmariae* seed stands; PH-A, Manchon del Abies; PH-R, Ranchito; PH-V, Ventana; PL-L, Laguna; PL-T, Tacuache.

Number in brackets, hybrid number at four of seven cone traits detected and at four of eight needle traits detected.

The results of the Principal Coordinates Analysis (PCoA) comparing genetic differences between individual specimens of PH and PL are shown in Fig. S2. At the individual level, the first three coordinates in PCoA explained 13.4% of the variability.

## Morphological detection of hybrids

*Pinus herrerae* and *P. luzmariae* are morphologically distinct and easily recognized by several needle traits as well as by the width, scale position and scale length of the cone. However, various morphological intermediates between the two species were found (Figs. 3 and 4), including hybrids confirmed by Random Forest (Table 5, Fig. 5) considering seven cone (hybrid proportion of 4.6%) and eight needle traits (4.5%). Every observation was correctly classified (TSS = +1). At the individual level, the first three coordinates in PCoA only explained 36.2% of the variability in seven cone traits, but 61.7% of the variability in eight needle traits (Figs. S3 and S4). Hybrids identified by 15 morphological traits matched only 13.4% of the molecularly detected hybrids, and 5.7% of hybrids were only found by morphological traits.

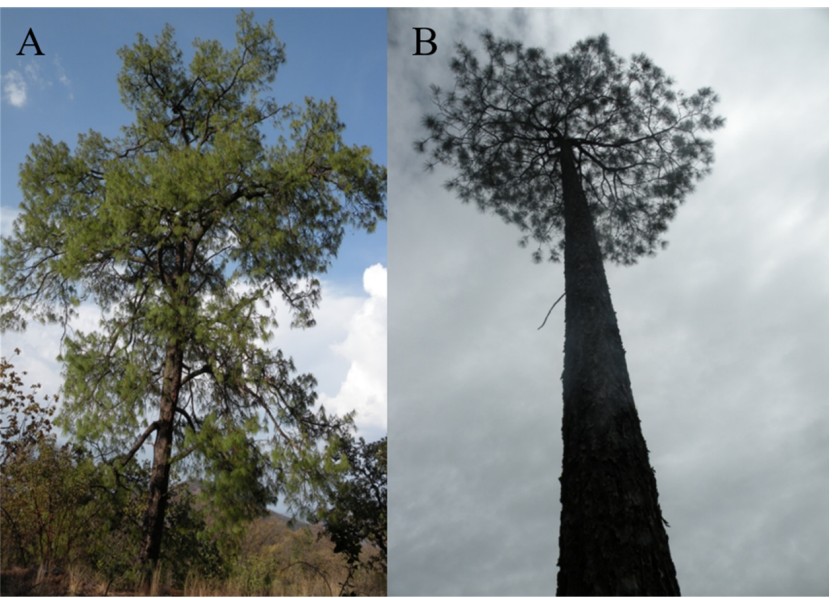

**Figure 3  Images from a typical *Pinus luzmariae* (A) and a *Pinus luzmariae* hybrid (B).** Images from a typical *Pinus luzmariae* (Bolaños in Jalisco, Mexico, 2013) (A) and a *Pinus luzmariae* hybrid (seed stand "Laguna" (PL-L), tree 102, 26 m stem height) (B).

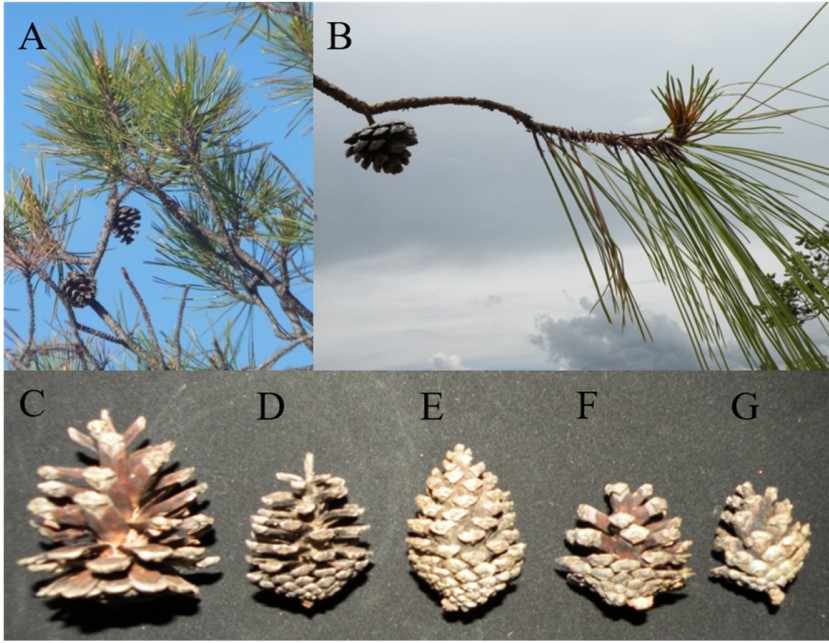

**Figure 4  Typical branches, needles and cones of *Pinus herrerae* and *Pinus luzmariae* and variation of *P. luzmariae* cones.** Typical branches, needles and cones of *Pinus herrerae* (A) and *Pinus luzmariae* (B) and variation of *P. luzmariae* cones: typical (C), different hybrid forms (D–G).

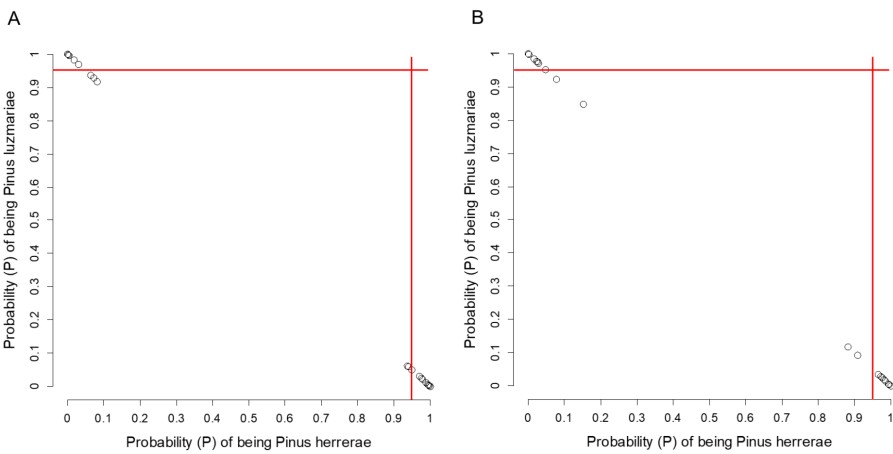

**Figure 5** **Posterior probability (*P*) of being *Pinus herrerae* and *Pinus luzmariae* using Random Forest classification.** Posterior probability (*P*) of being *Pinus herrerae* (PH) and *Pinus luzmariae* (PL) that a tree belongs to a particular class (PH or PL) using a Random Forest classification and (A) using seven cone traits, (B) using eight needle traits; True Skill Statistic (*Allouche, Tsoar & Kadmon, 2006*) = +1. If the *P* of PH (or PL) affiliation of a possible PH (or PL) tree was less than 0.95 (red lines), then, that tree was considered as a putative PH (or PL) hybrid.

## Clues of possible hybrid vigour (heterosis) in *P. luzmariae*

In this study of 69 *P. luzmariae* (pure and hybrid) trees, the hybrid heights and DBHs were much heterogeneously distributed than the dimensions of the pure trees. The smallest (one tree with 14 m height) and the tallest trees (14 trees with 23–30 m) were hybrids. The pure trees presented a normal distribution (probability) in which the expected proportion of trees higher than 24 m was much lower than the observed frequency of the tallest hybrids (Fig. 6).

## DISCUSSION

Species crossability in pines is of great theoretical and practical interest (*Lopez et al., 2018*; *Vasilyeva & Goroshkevich, 2018*). Many pine hybrids, including several Mexican species, have been planted in trials across southern Africa in different conditions and climate regimes (*Hongwane et al., 2018*). Here, we report for the first time about the occurrence of hybrids in *Pinus luzmariae*, a little known species, introgressed by *P. herrerae*, revealing taller trees in comparison to all populations previously known for the species (as compared with those described in *Pérez de la Rosa, 1998* and *García-Arévalo & González-Elizondo, 2003*). Populations of the introgressed *P. luzmariae* include trees 14 to 30 m (vs. 6-12 m in most other populations) (Fig. 6). This can be interpreted as hybrid vigour or heterosis, being the first report for Mexican pines. Other studies have shown that hybrid pines in the country do not differ from the pure trees in relation to vigour or robustness, e.g., *Pinus oocarpa × P. pringei* (*López-Upton et al., 2001*) and *P. arizonica × P. engelmannii* (*Ávila Flores et al., 2016*). In comparison with other populations of the same species, populations of the introgressed *P. luzmariae* (PL) display some important characters that are used to select superior forest trees (*Kedharnath, 1984*), i.e., good growth vigour, superior height,

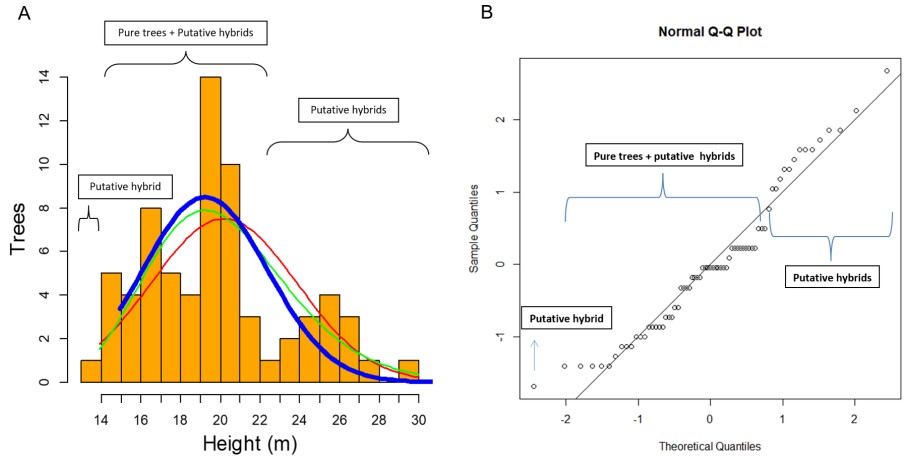

**Figure 6 Clues of possible hybrid vigour in *Pinus luzmariae*.** Clues of possible hybrid vigour in *Pinus luzmariae*: (A) Histogram (tree number) of tree heights (m), red line = normal distribution (probability) of all (69) pure *P. luzmariae* trees and putative hybrids under study, green line = logarithm normal distribution of all pure *P. luzmariae* trees and putative hybrids under study, blue bold line = normal distribution of the pure *P. luzmariae* trees under study, (B) normal Q-Q-plot of all (69) pure *P. luzmariae* trees and putative hybrids under study.

good self-pruning, and straight cylindrical bole. In the two introgressed populations, *P. luzmariae* formed almost pure, relatively dense stands with a few specimens of *P. herrerae* (PH) and *Pinus devoniana* in association, in contrast to the open, mixed stands in which *P. luzmariae* usually grows.

The high percentage of AFLP fragments (87%) shared by *P. herrerae* and *P. luzmariae* resulted in a large proportion of putative hybrids (54% by STRUCTURE (STR) and 38% by NewHybrids (NH)) using a posterior probability of at least 95%. The accuracy test detecting the different hybrid classes showed comparable results to other studies. However, NH detected more of both, accurate hybrids and individuals of pure species, than STR (Table 4). This explains the notable difference in the putative hybrid number found between STR (92 hybrids) and NH (65 hybrids). Therefore, the results presented by NH are probably more precise.

Previous studies have reported that the accuracy of these software can differ greatly depending on the population context (*Vähä & Primmer, 2006*). Using AFLP and a PP of at least 90%, *LaRue, Grimm & Thum (2013)* presented 100% accuracy rate of simulated $F_1$ hybrids, but lower percentages of $F_2$ and backcrosses (about 91% and 92%, respectively). *Cullingham et al. (2011)* found a mean power of 74% to detect hybrids using microsatellites and a PP of at least 90% for $F_1$ hybrids and PP < 90% for other hybrid classes.

The high degree of introgression can be explained by the relatively recent diversification of species in the subsection *Australes* and the very weak reproductive barriers between them (*Little & Righter, 1965*; *Garrett, 1979*; *Dvorak et al., 2000*; *Vargas-Mendoza et al., 2011*; *Gernandt et al., 2018*). Similar weak reproductive barriers and high recent speciation rates have been recorded for madrones (*Arbutus* spp.) and oaks (*Quercus* spp.) (*González-Elizondo, González-Elizondo & Sørensen, 2012*; *González-Elizondo, González-Elizondo &*

*Zamudio, 2012*; *González-Elizondo et al., 2013*; *Hipp et al., 2019*), the other two tree genera that, along with pines, dominate in the temperate forests of the Sierra Madre Occidental in western Mexico, where this study was carried out. Introgressive hybridization, although usually not obvious, may be more important in evolution than those cases in which hybridization is evident (*Anderson, 1949*; *Hipp et al., 2019*).

The interspecific gene transfer between the two pine species studied here is also supported by (i) wind pollination, (ii) longevity of individual trees, (iii) overlapping generations, (iv) large effective population sizes, and (v) weak physical barriers caused by sympatric distribution (*Ávila Flores et al., 2016*). The relatively high diversity and high levels of gene flow in trees (relative to herbs and shrubs) is favoured by their outcrossed mating system and long distance seed dispersal (*Petit & Hampe, 2006*).

Of the five seed stands studied, PL-L displayed the highest degree of hybridization (94%), confirmed by AFLP as well as cone and needle traits (Tables 3 and 4). According to the PCoA results (Figs. S2, S3 and S4), many PL-L individuals were genetically closely related to *P. herrerae* individuals. The high phenotypic plasticity and more luxuriant growth found in both populations of *P. luzmariae* under study (Tables 3 and 5) are a consequence of the hybrid origin. Crossbreeding or heterozygosity promotes variability, as found by *Strauss (1987)* for heterozygous trees of *Pinus attenuata* Lemm. derived from crossbreds. Resistance to disease, pathogens or environmental stresses has been a target in tree breeding towards interspecific hybrids. For example, *Pinus patula* Schltdl. & Cham. has been crossed with *P. tecunumanii* F.Schwerdtf. ex Eguiluz et J.P.Perry and with *P. oocarpa* in plantations in South Africa to increase tolerance to a fungal pathogen. The resulting hybrids of these three Mexican pines have a low frost tolerance, so new crosses were made until the finding that *P. patula* × *P. tecunumanii* from high elevations has a higher frost tolerance than *P. patula* × *P. tecunumanii* from lower elevations (*Mabaso, Ham & Nel, 2019*).

The high degrees of hybridization have several possible consequences: (i) extinction of one of the PH or PL parental species due to wasted mating effort or genetic swamping; (ii) reinforcement of species boundaries; (iii) creation of a third, hybrid species; (iv) formation of a stable hybrid zone; and (v) partial introgression between the two hybridizing lineages (*Chunco, 2014*).

The stands of *P. luzmariae* we studied appear to represent a stable hybrid zone, like the *Pinus engelmannii* stand reported by *Ávila Flores et al. (2016)*. This conclusion is supported by the fact that the *P. luzmariae* population displays higher fitness than other populations of the species. Hybrid speciation does not occur in the studied populations as the hybrids are not spatially or ecologically isolated from the parental species, and no novel variants of morphological traits were found (*Ungerer et al., 1998*). The PH-A displayed the highest degree of hybridization in three PH studied (at least 37.1% consisting of at least 25.4% trees detected by NewHybrids and four extra trees detected by the cone and needle traits) (Tables 3 and 5). PH-A was located next to the two *P. luzmariae* seed stands (PL-L and PL-T), and it is expected that it intercepts large amounts of *P. luzmariae* pollen. We can, therefore, conclude that gene flow has occurred in both directions, from PL to PH and vice versa. However, gene flow from PH to PL seems to be more effective as more hybrids were

found in the PL stands, both of which are located at lower elevations than the PH stands (Table 1).

Despite the large number of hybrids detected in the studied stands, the frequency of first-generation ($F_1$) hybrids and backcrossing was low (Table 3), indicating that hybrid crossing was usual in the seed stands. Similar results have recently been reported in species of *Salix* and for pines in the subsection *Ponderosae* and (*Fogelqvist et al., 2015*; *Ávila Flores et al., 2016*).

Natural hybridization has also been observed in other Mexican pine species (*Gernandt et al., 2018*). Previous studies of the subsection *Australes* identified natural hybridization between *P. oocarpa* × *Pinus caribaea* and *P. oocarpa* × *Pinus pringlei* only by morphological traits (*Styles & Stead, 1982*; *López-Upton et al., 2001*). In a study of Mexican pine species of the subsection *Ponderosae*, *Ávila Flores et al. (2016)* observed a high degree of introgressive hybridization between *P. engelmannii*, *Pinus arizonica*, *Pinus cooperi* and *Pinus durangensis*. AFLP markers detected most of the putative hybrids (58%), and only a few were detected by morphological features (15%). Hybridization was not detected by morphological traits in 74% of all hybrids detected by AFLP. Hybrids and backcrossing were also found in Mexican *Arbutus* species that are common in disturbed areas (*González-Elizondo, González-Elizondo & Sørensen, 2012*; *González-Elizondo, González-Elizondo & Zamudio, 2012*). Natural pairwise and triple hybrids have also been detected in numerous Mexican *Quercus* stands (e.g., *Peñaloza Ramírez et al., 2010*; *Hipp et al., 2019*). Natural hybridization between different *Populus* species and gene flow between cultivated poplars and native poplar populations have been described for European riparian forests and stands (e.g., *Van den Broeck et al., 2004*; *Smulders et al., 2008*; *Lexer et al., 2005*).

## CONCLUSIONS

Hybridization between *Pinus herrerae* and *P. luzmariae* in seed stands in the Sierra Madre Occidental of Mexico has occurred in both directions to different degrees. Estimates of the success of hybrid individuals may be biased in this study by the fact that sampling was conducted in seed stands (in which plus trees predominate). Further research is necessary to increase our understanding of how hybridization may influence silvicultural traits in Mexican pines, as well as their evolution and adaptation to climate change. The successful survival and reproduction of these hybrids over generations will depend on their attributes, their fitness and the environmental factors influencing them (*Strauss, 1987*), given that hybridization leads to individuals which widely vary depending of the context, location and involved species (*Gompert & Buerkle, 2016*).

We conclude that both morphological and molecular approaches are essential to confirm the genetic identity of forest reproductive material as PH and PL frequently hybridize in all seed stands under study. Such information is very important for developing effective future breeding programs and successful establishment of plantations (*Ávila Flores et al., 2016*; *Pérez-Luna et al., 2020*) as well as for improving planning of the management of natural stands.

Introgressive hybridization in seed stands of *Pinus herrerae* and *Pinus luzmariae* generated outstanding plus trees. Because of their tall, straight trunks, hybrids of the

largely unknown *Pinus luzmariae* represent a promising, valuable source of timber for wood industries as well as for reforestation in poor sites. The hybrid trees may be able to be cultivated after evaluation germplasm and vegetative propagation potential and may be suitable for commercial exploitation. However, further research is needed to examine the performance of hybrids and to assess their fertility and growth relative to those of pure species. Finally, monitoring natural hybridization is important in relation to sustainable forest management in Mexico.

## ACKNOWLEDGEMENTS

We thank Sergio Simental-Rodriguez, Saskia Friedrich and Javier Hernández-Velazco for assistance with fieldwork and dataset preparation and Imelda Flores for assistance with the morphometric study. This research study was conducted when the last author was visiting Instituto de Silvicultura e Industria de la Madera (ISIMA) at Universidad Juárez del Estado de Durango (UJED), Mexico, invited by Dr. Christian Wehenkel.

### Funding

This work was supported by CONAFOR, Mexico. The funders had no role in study design, data collection and analysis, decision to publish, or preparation of the manuscript.

### Grant Disclosures

The following grant information was disclosed by the authors:
CONAFOR, Mexico.

### Competing Interests

Christian Wehenkel is an Academic Editor for PeerJ.

### Author Contributions

- Christian Wehenkel conceived and designed the experiments, performed the experiments, analyzed the data, prepared figures and/or tables, authored or reviewed drafts of the paper, and approved the final draft.
- Samantha R. Mariscal-Lucero analyzed the data, authored or reviewed drafts of the paper, and approved the final draft.
- M. Socorro González-Elizondo and Matthias Fladung performed the experiments, analyzed the data, authored or reviewed drafts of the paper, and approved the final draft.
- Víctor A. Aguirre-Galindo performed the experiments, prepared figures and/or tables, and approved the final draft.
- Carlos A. López-Sánchez analyzed the data, prepared figures and/or tables, and approved the final draft.

### Field Study Permissions

The following information was supplied relating to field study approvals (i.e., approving body and any reference numbers):

A field permit was not required. However, we had a permit from the SEMARNAT to collect plant material in Mexico. We have included in the text: (collection permit SEMARNAT SGPA/DGVS/003644/18).

## Data Availability

Raw data is available in the Supplemental Materials.

## Supplemental Information

Supplemental information for this article can be found online at http://dx.doi.org/10.7717/peerj.8648#supplemental-information.

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
