# Peer review of "Tall Pinus luzmariae trees with genes from P. herrerae"

_PeerJ, doi:10.7717/peerj.8648_

## Round 0.1 · original submission · Major Revisions

Your manuscript is written in clear and unambiguous language and conforms to the standards in the field. Minor errors were found by the 3 reviewers. A major issue is in identification of pure and hybrid individuals. The procedure should be carefully described and (possibly) revised (see suggestions from Rev. 3). It is also important to include references to recent papers on the topic.

Reviewer 1 ·

Basic reporting

See below

Experimental design

See below

Validity of the findings

See below

Additional comments

The article is interesting, important and should be published with minor changes:

1: Title, why "... - an unexpected result..."? Maybe a more neutral title is the better choice.
2: Line 98 and 106 (447): Concle & Critchfield, (1998) correct: (1988).
3: Line 100 and 333 (544): Is that Little & Francis, 1965 or Little & Righter, 1965 ?
4: Line 152: (Fig. 2) correct to (Fig. 1).
5: Line 223: Include reference to (Fig. 2).
6: Line 387 (539) Lexer et al. 2008 correct: Lexer et al. 2005.

Reviewer 2 ·

Basic reporting

The paper is aimed to investigate the degree of introgressive hybridization between two pine species Pinus herrerae and P. luzmariae using a combination of phenotypic and AFLP-derived molecular data. Based on the assumption that individuals clustering with the majority of ones from another species are likely to contain a mixture of genetic information from both species, the authors identified the percentage of putative hybrids in four populations of P. herrerae and P. luzmariae. The analysis indicated that the hybridization occurred in all stands under study and that hybrids in P. luzmariae populations showed enhanced silvicultural traits compared to either parent species.

The paper provides appropriate background, describing the phenomenon of heterosis and briefly outlining the current state of affairs in studying population relationships among Mexican pine species. Previous works on using AFLP markers in detecting introgressive hybridization in plants are given, and the advantages of this method are described. However, the weaknesses of using AFLPs in population genetics are not mentioned.

The manuscript is written in clear and unambiguous language and conforms to the standards in the field. Although, there are some minor corrections that should be made in order to improve the text:
1. In Line 157 the authors describe the location of the tree stands and mention distance between study sites but refer to Fig.2 which depicts the results of clustering based on AFLP data. The reason for such reference in section ‘Study sites’ is not clear.
2. The term ‘Dasometric’ in Line 176 rarely occurs in literature. This word seems to be a direct translation of Spanish term ‘dasometría’ which implies the application of statistical methods for forest measurements and, to the best of my knowledge, is not used in international scholarly publishing. I would recommend that the authors consider using more common terminology to avoid confusing readers.
3. It seems that there are missing words in Lines 200-201. Is it 16 randomly chosen individuals from each population?
4. In Lines 279-280 PH stands are called red and PL blue, whereas in Figure 2 (which these lines are referring to) PH are blue and PL are orange.
5. The second paragraph in the Conclusion section contains repetition of ‘essential’ in Lines 398 and 400.

Comments on Figures and Tables:
1. There are some inconsistencies in Figures labeling. For instance, in Figure 1 populations of P. luzmariae are marked with red triangles and P. herrerae with yellow circles, while the caption says different. In Figures 2, 3, 6 and 7 there is a population marked as PH-T which is not described in Study sites section (I assume this is a typo).
2. Additionally, Figures 3, 4, 5, 6 and 7 consist of two or more parts but do not have proper labels such as “A, B, etc.”. The authors should ensure that all the figures have the correct labeling as described in the Submission Guidelines.
3. In my view, Figures with PCoA clustering results (Fig. 3, 6 and 7) would benefit from being colored and uniform in size and quality. As well as adding labels to the photographs of trees and cones (Fig. 4 and 5) identifying which species or hybrid they belong to would improve clarity.
4. Finally, it seems like a good idea to represent the last paragraph in the Study sites section as a table or add this information to the one that describes the locations of the stands.

Experimental design

A great amount has been conducted by the authors to collect the specimens and morphological data across several seed stands of Sierra Madre Occidental. The methods used in the paper conform to the stated question of identifying hybrids between two close species. The authors use established a protocol for performing AFLP analysis, and traditional approaches to processing the data. However, the procedure of testing the affiliation of individuals to each species using Bayesian clustering is not clear. Particularly in Line 225 what is meant by “association between PH and PL trees”?

The use of Affinity Propagation (AP) clustering raises some questions. It is said that “AP has several advantages over related cluster techniques”, but these advantages are not listed. Furthermore, AP does not require a predefined cluster number, however, in the case of identifying hybrids between two species the number of clusters is known. Hence the advantage of using AP over k-means is not obvious. I would suggest authors including more information that clarifies their choice of this method.

Additionally, it is not stated in the text how the authors identified the hybrids with the PCoA clustering, i.e. what thresholds (if any) they use to detect hybrid individuals.

Validity of the findings

My major concern is related to the lack of the exact numbers in the description of the clusterization results. The authors present only proportions of identified hybrids (Tables 3 and 4) not providing precise numbers of putative hybrid individuals (although some numbers appear in the text) detected by processing molecular or morphological data.

Discussing the results authors indicate that “gene flow from PH to PL seems to be more effective as more hybrids were found in the PL stands”. How this can be explained (for instance, it can be noticed that stands of P. luzmariae are generally located lower compared to P. herrerae stands)? I would suggest elaborating on this a bit more.

As a minor comment regarding data representation, “apcluster” package has built-in plotting tools so the results of clusterization can be easily visualized and presented in a more reader-friendly way.

Reviewer 3 ·

Basic reporting

• Authors present the work in clear professional Engilsh
• Introduction provides appropriate context.
• Literature is somewhat dated – newer references would be appropriate in many places throughout the intro
• Figures are reasonable, but some suggested changes are provided below
• Raw data are provided, although one Table is missing (see below)
• The Methods are lacking several critical pieces, which make it difficult to assess the validity of the Results.

Experimental design

• This work is within the scope of the journal
• Research question is well defined and fills a knowledge gap
• I have identified several issues that the authors must address to ensure the validity of the work
• Additional information and clarification is needed within the Methods

The authors did not clearly define the pure individuals that serve as references to define the hybrid nature of each individual in the study. This is a critical step for any project with this scope. I am hoping that this was an oversight. If pure individuals do not exist, then I would consider this a fatal flaw in the design of the study.

With pure individuals it is possible to assess the power to identify hybrid individuals within the five populations examined by the authors.

I need further information to fully assess the strength and design of this study.

Validity of the findings

• To assess the validity of the findings, there are issues that need to be addressed, in particular the presence of pure individuals to use as references for establishing the diagnostic nature of the molecular markers

• Underlying data are provided, with the exception of one critical table (individual collection data)

Additional comments

Detailed Comments
The authors present an interesting study examining the potential impact of hybridization on pine tree growth in Mexico. My biggest concern is a lack of pure reference populations needed to define the diagnostic state of the difference AFLP fragments used to identify hybrids. I hope that this was an oversight by the authors and this can be clarified. I outline additional issues below.

Introduction
• Throughout, there are mostly older references - please try to identify newer references that discuss issues such as hybridization
• Line 74-96: This section seems out of place relative to the surrounding text. I suggest a reorganization
• Line135: The topic sentence of this paragraph needs revision to clearly introduce this paragraph.

Methods
The authors need to provide greater information in the section for the Molecular Detection of Hybrids. First, do they include individuals that are known pure species? This to detecting portions of the genome that are introgressed into the other species. Inclusion of these types of samples is considered standard practice. Second, the descriptions of the analytical methods used in this section require greater detail. For example what were the protocols used for STRUCTURE (e.g. burnin, # generations)? Third, I am concerned about how hybrids are defined in this context. I would strongly recommend a more quantitative approach (For example: Cullingham et al. 2011
https://doi.org/10.1111/j.1365-294X.2011.05086.x)

I would also like to see a figure that shows the morphological measurements that were taken on the cones and needles. While this information may be present in an earlier paper, it is important that this manuscript is able to stand independently. Clear descriptions of the morphological traits and a figure that shows each would be needed.

• L199: Clarify your meaning of reference samples
• L230: What do you mean by putative pure individuals?
• L248: What are the voucher numbers for the individuals deposited in to the collection?
• L249: Write out the name of CIIDIR in full

Results
• The authors need to provide a Table that includes the collection data and assignment of each individual. It should also include the voucher number for each specimen and any other associated data (individual NewHybrid results, structure assignment).
• L279: There is confusion here with the colours used to denote PH and PL. Here the authors state that PH has a dominante variant (Red). In the Structure plot, PH = blue. Where does the red come from?
• Line297: What is your power to detect hybrids? See Cullingham et al. 2011 for methods to determine your to detect hybrids
Discussion
• Line317: The first sentence of the Discussion mentions hybrid vigour. There were no data that measure hybrid vigour in the trees tested for introgressed alleles. There should be a section identifying traits that define hybrid vigour (tree height, straightness) for each individual tree measured for the study.
• Line 330: How can you be certain that these shared fragments are due to introgression rather than shared regions due to a shared anscestory? You would need to have pure individuals of each species to define what are Ph and Pl markers prior to identifying hybrids
Table 2
• I think it would make more sense for this table to show the differences between the two species (i.e. the species diagnostic traits) as well as the ranges for the populations that are being studies. This speaks to the larger issue of not having clearly defined pure individuals/populations.
Figure 2
• Identify the different hybrid classes for each individual within this Structure plot
Figure 3
• Difficult to differentiate the PH-T and PH-V triangles. Revise.

Figure 5
• Label identity on each figure panel

---

## Round 0.2 · Minor Revisions

Thank you very much for greatly improving the manuscript. There are several remaining items. First, it is important to make all raw data available and provide clear instructions for the readers about the access.
Second, additional clarification of the LDA is needed (see comments from the second reviewers).

Finally, there are several typos that need fixing.

Reviewer 1 ·

Basic reporting

.

Experimental design

.

Validity of the findings

.

Reviewer 2 ·

Basic reporting

The article is written in clear language and all appropriate background is provided. The authors substantially revised the manuscript, adding more recent literature and changing some of the used methods (see comments below).

Experimental design

There are a couple of questions regarding the description of methods:
• Was the distribution of morphological traits tested for normality before performing LDA? It may be worth explicitly specifying what methods were used (graphical or some frequentist test).
• Did you use the training set for LDA to get LD functions? Also, was any standardization used for the variables? You may wish to clarify this in the methods section to ensure reproducibility.

Validity of the findings

See below

Additional comments

There are some minor notes on the changed parts of the manuscript:
• Lines 237-238, the end of the sentence looks a bit strange, it might be worth revising
• Line 300, probably a missing preposition
• Line 404, the word “respectively” seems to be out of place
• In Figure 6, does “the one linear discriminant” in captions mean the first discriminant function?
• In Figure 7 the left tail (one tree with 14 m height) is not shown in the histogram, although it is mentioned in the text.

Reviewer 3 ·

Basic reporting

The authors use clear language throughout the manuscript. They have updated their references to provide more current background/context.

I find no evidence that the authors have shared the raw AFLP data or morphometric data set.

Experimental design

My biggest concern in an earlier version of the manuscript was that the authors had not defined "pure" species in their data set. They have now been clearer in their revisions about how they identify "pure" individuals. Ideally, I would prefer to have the pure individuals identified a priori, rather than analytically after the fact, but I acknowledge that this is not always feasible. As such, I am okay with the current revisions.

Validity of the findings

With the improvements on pure vs hybrid differentiation, I believe that the authors have provided evidence to support their conclusions.

Additional comments

Thank you for your attention to details and efforts to revise the manuscript in response to my earlier comments.

Please indicate where you will deposit your raw data.

---

## Round 0.3 · accepted · Accept

The paper was improved. Although AFLP technology used in the paper has its limitations, you have described the advantages and shortcomings of the approach, so the readers are not misled. I recommend the manuscript for publication